# A Large-Scale Validation of an Improved Embryo-Rescue Protocol for the Obtainment of New Table-Grape Seedless Genotypes

**DOI:** 10.3390/plants12193469

**Published:** 2023-10-03

**Authors:** Emanuele Chiaromonte, Giovanna Bottalico, Pierfederico Lanotte, Antonia Campanale, Vito Montilon, Massimo Morano, Antonia Saponari, Costantino Silvio Pirolo, Donato Gerin, Francesco Faretra, Stefania Pollastro, Vito Nicola Savino

**Affiliations:** 1Department of Soil, Plant and Food Sciences, University of Bari, Via G. Amendola 165/A, 70126 Bari, Italy; emanuele.chiaromonte@uniba.it (E.C.); vito.montilon@uniba.it (V.M.); donato.gerin@uniba.it (D.G.); francesco.faretra@uniba.it (F.F.); stefania.pollastro@uniba.it (S.P.); 2Institute for Sustainable Plant Protection–Support Unit Bari, National Research Council of Italy (NRC), Via G. Amendola 122/D, 70126 Bari, Italy; pierfederico.lanotte@ipsp.cnr.it (P.L.); antonia.campanale@ipsp.cnr.it (A.C.); 3Italian Variety Club, Via Cisternino, 281 c/o CRSFA Basile Caramia, 70015 Locorotondo, Italy; massimomorano84@gmail.com; 4CRSFA—Centro Ricerca, Sperimentazione e Formazione in Agricoltura, “Basile Caramia”, Via Cisternino 281, 70010 Locorotondo, Italy; antonellasaponari@crsfa.it (A.S.); viton.savino@gmail.com (V.N.S.); 5Servizi Avanzati per la Sostenibilità e l’Innovazione nelle Aree Agricole e Rurali Sinagri S.r.l., Via G. Amendola 165/A, 70126 Bari, Italy; costantino.pirolo@gmail.com

**Keywords:** embryo rescue, grapevine, crosses, culture medium, sampling time

## Abstract

The new trends in the consumption of table grapes and the growing interest in the environmental impact of this crop have pushed breeders toward the development of seedless cultivars endowed with resistance, through crossbreeding programs. To obtain seedless grapes, the use of embryo-rescue techniques is fundamental. In this research, a grape embryo-culture protocol was optimized and validated by using 39 cultivars and 41 cross-combinations carried out in the framework of a large private table grape program of the private network Italian Variety Club in the period 2017–2021 evaluating several factors, such as the improvement in embryo formation, germination and growth, and plantlet development. The embryo culture attitude of crosses between different combinations of seedless parents was assessed, and the rates of embryo development from the extracted ovules mostly ranged from 3.5 to 35.5% with 5 out of 43 genotypes outliers. Experiments conducted at different sampling times, in a range of 43–62 days after pollination (DAP), did not show significant differences between the samples analyzed, while the rate of embryos developed with the applied protocol proved its employability on multiple genotypes, although the grapevine genotype significantly influenced the technique efficiency.

## 1. Introduction

The European Union is the world’s main grape producer, with an estimated production of 1.7 million tonnes of grapes for fresh consumption in 2021 [1]. The market of table grapes is the most important, with an import value of EUR 1.6 billion, after bananas and avocados, in the European scenario. With a production stabilized around 1 million tons per year of table grapes, Italy is the main producer in Europe [1]. In addition to volume, the strong point of Italy is the extensive area of cultivation of organic grapes (2177 hectares of organic table grapes in 2017) [2].

To improve production, Italian growers are slowly joining the seedless trend, mostly in the Puglia region, and at the same time, they are looking for late cultivars to extend the seasonal availability of produce. In the development of new cultivars, the main sought characteristics are the general berry traits (weight, color, shape, and skin thickness), flavor (aromas and sweetness), fruit shelf life, harvesting time, and tolerance to pathogens, such as *Plasmopara viticola* and *Erysiphe necator*. However, nowadays, the seedlessness and disease resistance traits are the most desirable [3]. Over the last few years, the quality and nutritional composition and the breeding of new seedless cultivars have been the most significant efforts in research [4,5,6].

From a botanical point of view, seedlessness is the phenomenon whereby fruit is formed without developing seeds, and plants are sterile due to the absence of embryos. Particularly in the cultivated grapevine (*Vitis vinifera* subsp. *sativa* L.), parthenocarpy and stenospermocarpy are the two types of fruit seedlessness [7,8]. In parthenocarpic cultivars (Corinth type), berry production is not preceded by ovule fecundation, while in stenospermocarpic cultivars (Sultanina type), some of the ovules possess normal embryo sacs, fertilization occurs, and embryos may develop, but their development is often stopped, and normal seeds cannot form [9,10]. Despite this information, nowadays, the exact causes of embryo abortion in seedless grapevines are still not fully understood; the main hypothesis proposes that parental tissues alter the hormonal balance during the first steps of the ontogenetical cycle, causing the embryo’s abortion [11]. It is then obvious that conventional breeding methods cannot be applied for genetic improvement, and in vitro techniques are the only available approach. The first reported vines originated via ovule culture from the seedless grape in 1983 [11], and it was followed by several other reports [12,13,14,15,16,17,18,19,20,21,22,23]; the exploited biotechnological principle consists of the cultivation of extracted ovules or immature embryos in artificial media, to admit the development of the plantlet without the abortion caused by a natural process. Nowadays, grape breeding projects using this in vitro technique can improve the efficiency, by reducing the time required to obtain seedless cultivars, usually from 6 to 8 years [20]. Moreover, by adopting seedless stenospermic instead of seeded cultivars as female parents, the frequency of seedlessness found in the progeny could be increased by up to 10–15% [24]. When the rescue is performed in “seedless × seedless” crosses, where female and male parents are from a “seeded × seedless” or a “seedless × seedless” cross, a high percentage of seedless progeny with natural big berries and small seed traces is obtained [3,25]. The main steps of the embryo-rescue technique are as follows: (i) in vitro culturing of ovules, (ii) embryo collection and culturing for plant development, and (iii) managing of rooted plantlets (elongation, acclimatization, and transplanting to soil) [24]. The most crucial step is the first because an optimal culture medium is the key to a successful rescue of potentially abortive hybrid embryos. Indeed, an appropriate composition of growth regulators in the medium is crucial for embryo germination and growth and plantlet development. However, the number of obtained embryos also depends on the parent’s genotype and harvesting [11,26,27,28,29,30,31,32,33]. On stenosternocarpic seedless grapes, collection time is important to block embryo abortion, considering that it determines the herbaceous or woody texture of seed tissues [11,31,33,34]. According to previous reports, the best sampling time to achieve the maximum efficiency of the embryo-rescue protocol is 40–60 days after pollination [14,33,35,36].

The technique of the rescue of immature embryos provides important genetic gains for the seedless feature, due to the size reduction of seed traces resulting from crosses between seedless parents [26]. The advantages of the technique were also evaluated in the genetic improvement of grapevine for disease resistance [37], a highly relevant issue since the 50% cut in pesticide use by 2030 is a key goal of the European Farm to Fork strategy. The aim of this work was the improvement and validation of an embryo-rescue protocol working with embryos from several crosses between seedless parents selected for features like moscato flavor and putative resistance to the most relevant pathogens, *E. necator* and *P. viticola*. 

## 2. Results

### 2.1. Efficiency of the Embryo-Rescue Protocol

Data on the mean numbers of berries and ovules collected from each bunch, the numbers of ovules extracted and those developed from each berry, and the proportions of developed embryos on the in vitro settled ovules are reported in Table 1.

In five years of work, a total of 43 crosses carried out with 25 female parental table-grape cultivars were compared. In detail, 449 bunches and 39,351 berries were manipulated to obtain 88,837 in vitro settled ovules and 11,941 germinated embryos. For each bunch, the number of berries was in the range of 27 to 215, and each berry had on average two ovules. Significant differences were recorded in the six crosses in which TS28, TS25, and TS27 were used as the female parent. The ratios of ovules:berries were below 1.5:1 in the four crosses including TS28 and TS25, while the ratio values were higher than 3.2:1 in the two crosses including TS27 as female parents. 

The percentage of ovules that developed embryos was in the range of 3.5% to 35.4% for 38 crosses, with 5 out of 43 outlier genotypes. The embryogenetic efficiency was indeed less than 3.5% in the cross TS04 × TS24 and all three crosses with TS25 as the female parent, while the cross TS29 × TS38 yielded the highest efficiency with a value of 41.1%. Summarizing the data from all the crosses, the percentages of ovules developed in embryos were less than 5.0% for only 2 crosses and were in the range of 6.0–10.0% for 11 crosses, 11.0–20.0% for 14 crosses, and higher than 20% for the remaining 8 crosses. The protocol efficiency, relative to the number of embryos developed by settled ovules, was between 3.0 and 14.0% in more than half of the crosses and between 8.2 and 11.8% in all the crosses involving TS27 as the female parent.

As expected, not all the obtained embryos were successful in developing plantlets (Table 2 and Figure 1). Different causes, including in vitro and in vivo microbial contaminations were responsible for losses during the acclimatization steps; the percentages of embryos that originated plants were in the range between 10.0% (TS15 × TS32) and 62.4% (TS33 × TS28), but only in six crosses were below 20.0%. More than half of the crosses had a percentage of losses in the stage of acclimatization below 60.0%. Considering the ovules, the percentage of plantlets was in the range of 0.3% (TS04 × TS24; TS25 × TS39) − 20.5% (TS29 × TS38). 

### 2.2. Sampling Time

Detailed results obtained by comparing cultivars characterized by a similar ripening time (grouped as middle, middle-late, and late ripening time) for the numbers of ovules extracted per berry and the percentages of settled ovules developed in acclimatized plants at different sampling times are reported in Appendix A and Figure 2.

Considering the crosses characterized by the middle ripening moment of the female parent, the mean ratio of ovules:berries showed a tendency to increase for the earlier three sampling times and decrease at 58–62 days after pollination (DAP), so that the highest values, over 2.5 ovules per berry, were recorded at 53–57 DAP. The cross TS27 × TS03 showed the highest outstanding value of 3.4 ovules:berries at 48–52 DAP. For the same cultivars, the embryogenic efficiency was between 10.0 and 12.0% for the earlier three sampling times, while the highest mean value (17.1%) was recorded at the last sampling. Almost all the data were in the range of 4.0–20.0%. The only exception was the cross TS17 × TS27 at 43–47 DAP (24.7%).

For the crosses with female parents with middle-late ripening time, the ratios of ovules:berries were in the range of 1.5 to 2.6. TS04 × TS24 was the only cross sampled at the first sampling time and yielded an ovules:berry ratio of 2.3 and a percentage of embryo development of 0.8%. The ovules:berries ratios showed the highest mean value (2.5) at 48–52 DAP; after that, the value decreased and increased again at 58–62 DAP (2.3). Ovules:berries ratios for all crosses were in the range of 1.5–2.6. The best embryogenic efficiency in this group was 20.1%, recorded at 53–57 DAP, while the highest value (27.6%) and the lowest value (1.0%) were recorded at 58–62 DAP (TS33 × TS28) and 53–57 DAP (TS04 × TS24), respectively. The sample at 53–57 DAP was characterized by two outstanding high values, 20.1% for TS23 × TS28 and 0.9% for TS04 × TS24.

For the late-ripening cultivars, only the cross TS09 × TS04 was assayed at 43–47 DAP, yielding an ovule:berry ratio of 2.5 and a percentage of embryo development of 7.0%. For the ovules:berries ratio, the highest (2.5, TS09 × TS10) and the lowest (1.2, TS25 × TS39) values were recorded at 48–52 DAP. As for the mean value for all the crosses, the highest value (2.2) was recorded at 53–57 DAP and the lowest value (1.7) at 58–62 DAP. In this group of cultivars, the protocol efficiency was between 0.9% (TS25 × TS13) and 22.4% (TS05 × TS27), respectively, at 48–52 DAP and 53–57 DAP. The mean values of embryogenic efficiency for all crosses were between 5.8% at 43–47 DAP and 3.4% at 58–62 DAP.

## 3. Discussion

Since the first report of the usage of in vitro ovule culture for originating recombinant seedless grape genotypes [11], the usefulness of the technique has become clear, and numerous factors have been studied to improve the efficiency of applied protocols. It was proved that the number of obtained embryos depends on the culture medium [29,33], parental genotypes [27,28], and the time of berry collection [11,13,29,30,32,38,39]. Several papers reported the use of Nitsch & Nitsch’s (NN) salts [15,20,40,41], different hormones, such as IAA or GA3 [38,42], as well as activated charcoal [15,43,44], and mashed banana [33]. Most of the factors were also studied in this research, paying particular attention to grapevine genotypes and the role of hormone concentrations in the culture medium [38]. The outcome was an improved medium for the stabilization of explanted ovules and for embryo growth. The medium contained macro- and NN microelements [45], IAA (3.00 mg/L), and GA3 (2.00 mg/L) and was supplemented with activated charcoal (2 g/L) for reducing tissue browning and embryo abortion rate [41,42]. The improved medium and the related protocol were validated over five years on 43 different crosses involving 39 grapevine genotypes characterized by different ripening times. Overall, the adopted protocol yielded an efficiency of embryos developed by in vitro settled ovules ranging from 3.0 to 14.0% in more than half of the assayed crosses.

The large-scale experiments on very high numbers of berries, ovules, and embryos allowed us to ascertain the adaptability of the newly developed protocol to different grapevine genotypes. The obtained efficiencies of plantlet formation were in line with previously reported data [27,33,34,38,41,46].

The results of this research evidenced a high variability of the ratios between embryo growth and ovules, in the range from 0.8% (TS04 × TS24) to 41.1% (TS29 × TS38), depending on the crossed grapevine genotypes. The influence of the genotype was also confirmed by the similar results obtained in different years using TS27 and TS25 as the female parent in crosses, with ratios around 10.0 and 2.0%, respectively. Our results align with the ones reported by Puglisi et al. [27] reporting great variability among parents, also when the same cultivar was used as the male or female parent.

The efficiency of the new protocol in terms of percentages of acclimatized ready-to-use plants from grown embryos was in the range of 10.0–62.0%, fitting well with the results obtained by Jiao et al. [28] reporting a plantlet development rate between 17 and 58%, using 15 different genotypes and 11 cross combinations [28].

An overall protocol efficiency over 20.0% was obtained by the cross between TS29 (female, seedless, early ripening) and TS38 (male, seedless, early ripening), confirming the influence of grapevine genotypes on the embryogenic ability of ovules, on the ability of plantlet development in vitro and, consequently, on their aptitude to breeding [18,27,28,31,36].

Li et al. [47] observed different abilities of embryo growth depending on the sampling time of berries in the cultivars “Thompson seedless”, “Flame seedless” (TS11), and “Ruby seedless”. They concluded that the length of time elapsing between pollination and berry collection was significantly and negatively correlated with the proportions of grown embryos in seedless grapes. Similar findings, but under a significant influence of grapevine genotypes, were also reported by others [30,31,33,48]. We applied the new protocol to berries collected at four five-day intervals in the period from 43 to 62 DAP. Overall, data showed a tendency for a higher protocol embryogenetic efficiency with sampling at 58–62 DAP. These results agree with findings reported by Guo et al. [48] based on observations carried out on crosses in which the seedless cultivars “Jumegui”, “Kyoho”, and “Red globe” were used as the female parent. Nevertheless, the embryogenetic efficiency did not show significant differences at different sampling times for crosses involving grapevine genotypes with medium, medium-late, or late ripening as also reported by Kebeli et al. [29]. These findings can be relevant for the application of the embryo-rescue protocol in large-scale breeding projects in which a very high number of berries need to be processed. The slight, if any, influence of sampling time on the results in terms of final plant production, indeed, could reduce the laboratory workload in the earlier and time-consuming steps of the protocol.

## 4. Materials and Methods

### 4.1. Embryo-Rescue Protocol

All the grape seeds used from the embryo rescue activities originated from crosses between different *V. vinifera* cultivars. The vines were grown in commercial vineyards located in Puglia (Southern Italy). Thirty-nine cross-combinations were set up during a period of five years (2017–2021). The parental genotypes herein used (Table 3) were selected by the Scientific Committee of the Rete Italian Variety Club, a network joining 23 private companies (www.reteivc.it (accessed on 30 March 2023)) [49], in the frame of a multi-year and extensive table large breeding program launched in 2014. Selection was mainly for the following characteristics: seedlessness, berry size, color, pulp firmness, ripening time, and putative resistance to *E. necator* and/or *P. viticola*. The cultivars used as parents are almost all patented so the lab code TS01–TS39 was used instead of the cultivar name, according to the confidentiality agreement between researchers and breeders. If known, their pedigree is reported. For the same reason, no details on the crosses are herein supplied.

Unlike a previous breeding work in which seedless cultivars were only used as pollen parents [50], the development of the ovule-culture protocol made it possible to use seedless vines as both parents [11,12,13] increasing the number of progenies marked by the seedless characteristic [51]. A traditional grapevine breeding technique based on the emasculation of flowers, as proposed by Eibach [48], was used. Briefly, the inflorescences of already fertilized flowers characterized by raised calipers were removed. At a different time during May, before blooming, flowers of the female parent of the cross were emasculated and closed in paper bags containing pollen from the male parent. Details on crosses are in Table 4.

Immature bunches were sampled 43–62 days after bloom. After harvesting, bunches were placed in a container and maintained at about 4 °C during the delivery to the laboratory where they were processed. Berries from each bunch were gently washed with soap under running water, and each berry was decontaminated in a solution of 1.4% sodium hypochlorite for 20 min under a laminar flow hood. To remove sodium hypochlorite residues, berries were washed twice with sterile distilled water.

Air-dried berries were dissected longitudinally with a sterile scalpel (Figure 3a) with the help of a stereo microscope under a laminar flow hood, and ovules, well cleaned by all berry tissue residues, were gently and accurately clamp-collected. The ovules were placed in Petri dishes (diameter 90 mm, height 16.2 mm) containing about 20 mL of the culture medium. Based on previous works [15,40,43,44,45], the composition of the medium, containing active charcoal, used in this study was optimized as reported in Appendix A. In each Petri dish, 30–40 explants from the same bunch were placed (Figure 3b). Plates, sealed with Parafilm, were maintained at 24 ± 3 °C under lighting from white light LED with a luminous intensity of ≃3000 lux, with a photoperiod of 16/8 h. From 8 to 30 weeks, the spontaneous growth of embryos (Figure 3c) was checked by the naked eye. After ≃10 weeks, the embryos were manually collected from ovules showing externally brown tissues (Figure 3d). Due to the long-time growth required, ovules were individually transferred onto a fresh growth medium, especially when dehydration or microbial contamination occurred. The grown sprouts were transferred into new sterile, transparent, and airtight 60 mL-sized containers (Figure 3e), containing ≃15 mL of a culture medium with no hormones for settling. Details on the used media are available in Appendix A [52,53].

The sprouts, properly oriented according to the embryo orientation, were inserted approximately 1 mm into the medium and maintained under the same above-described growing conditions until the complete plantlet development (Figure 3e). Once the plants within the substrate developed their first two true leaves and reached a height of at least 3–4 cm, they were ready for transplanting. Plantlets were individually collected from in vitro containers and transferred to previously rehydrated disks of dehydrated peat (Jiffy-7 of 44 mm, Jiffy Products International AS, Norway). Properly carded plants were placed in trays to facilitate their watering. Trays, covered with a transparent plastic film to ensure a gradual transition to the external conditions, were maintained at 24 ± 3 °C in a climatic chamber equipped with LED lights (AGRO light of 22 W) and a photoperiod of 16/8 h. About 10 days later, the plastic film was gradually removed. Well-developed plants were finally transferred into pots (9 cm × 9 cm × 10 cm) containing peat and maintained in the growth chamber until reaching a height of about 12 cm, at which point, they were transferred to the screenhouse (Figure 3f).

### 4.2. Influence of Sampling Time

For some crosses, bunches were sampled at two to three time points in the range 43–62 DAP to evaluate the possible influence of the female parent genotype on the best collection time [13,30,31]. The sampling times were classified into four groups each of 5 days: 43–47, 48–52, 53–57, and 58–62 DAP (Table 5). Ovules were processed as described above.

### 4.3. Statistical Analysis

For each experiment, numbers of ovules extracted from each berry, embryo growth rates, and percentages of ovules yielding acclimatized plants were determined as follows:-Number of ovules extracted from each berry = total number of ovules extracted and settled in vitro/total number of sampled berries.-Embryo germination rate (%) = number of embryos grown from in vitro settled ovules ×100/number of in vitro settled ovules.-Acclimatized plant rate (%) = number of plants acclimatized from in vitro settled embryos ×100/number of in vitro settled ovules.

Single bunches were considered biological replicates when appropriate, and standard error was calculated per each cross.

For the statistical analysis of the data, Microsoft Excel of Microsoft 365 was used.

## 5. Conclusions

A new improved embryo-rescue protocol for the obtainment of new seedless grapevine genotypes was validated in a large-scale experiment on a set of 43 crosses involving 39 parental seedless genotypes over 5 years. The feasibility of the new protocol was proved for all parental genotypes although these influenced the development rates of in vitro settled ovules. The sampling time of grapes in vineyards was evaluated in the range from 43 to 62 days after pollination, and it had little influence on the protocol efficiency, although better results were generally obtained when it was carried out from 53 to 62 days after pollination. The little influence of sampling time could be due to the main usage of crosses involving middle- to late-ripening cultivars. Research on this hypothesis is in progress. The obtained results highlight the relevance of a careful selection of parental genotypes in crosses by breeders due to their influence on the success of the economic investment required for the genetic improvement of the grapevine.

## Figures and Tables

**Figure 1 plants-12-03469-f001:**
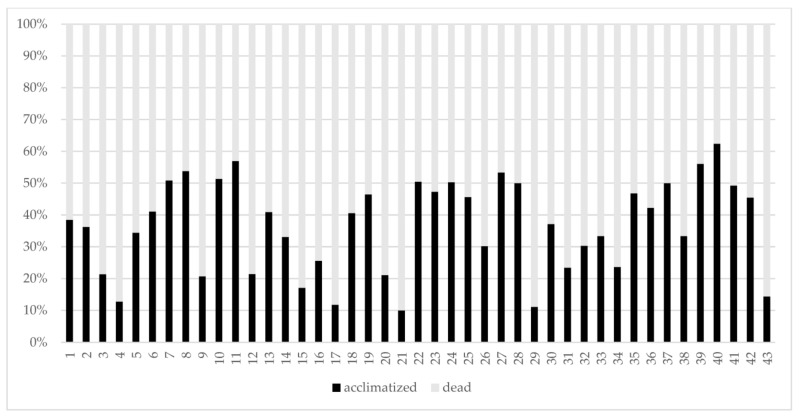
Percentages of plantlets developed from in vitro settled embryos. Cultivars employed in each cross-combination: 1: TS01 × TS04; 2: TS02× TS04; 3: TS04 × TS19; 4: TS04 × TS20; 5: TS04 × TS24; 6: TS05 × TS27; 7: TS06 × TS20; 8: TS06 × TS24; 9: TS07 × TS08; 10: TS09 × TS04; 11: TS09 × TS10; 12: TS09 × TS13; 13: TS09 × TS34; 14: TS11 × TS24; 15: TS12 × TS24; 16: TS13 × TS24; 17: TS13 × TS24; 18: TS13 × TS27; 19: TS14 × TS21; 20: TS15 × TS13; 21: TS15 × TS32; 22: TS17 × TS27; 23: TS22 × TS16; 24: TS23 × TS24; 25: TS23 × TS28; 26: TS24 × TS13; 27: TS25 × TS13; 28: TS25 × TS18; 29: TS25 × TS39; 30: TS27 × TS03; 31: TS27 × TS21; 32: TS27 × TS31; 33: TS27 × TS32; 34: TS27 × TS32; 35: TS28 × TS04; 36: TS28 × TS13; 37: TS29 × TS38; 38: TS30 × TS13; 39: TS33 × TS19; 40: TS33 × TS28; 41: TS35 × TS03; 42: TS36 × TS26; 43: TS37 × TS19.

**Figure 2 plants-12-03469-f002:**
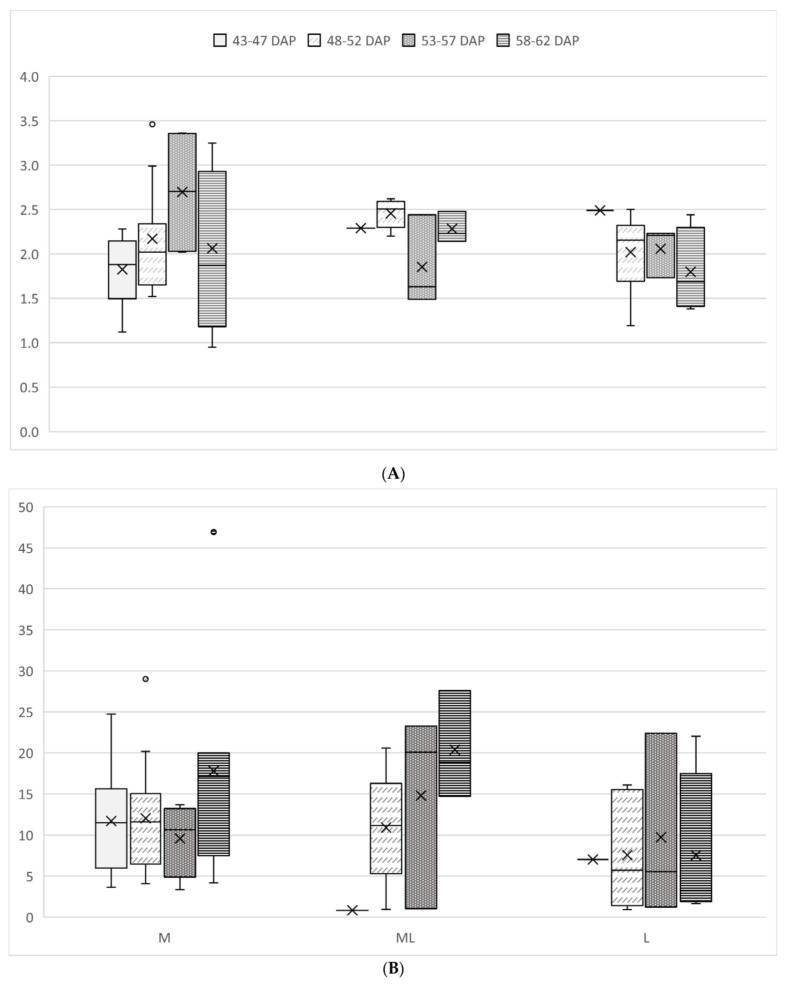
Comparison of ratio ovules:berries (**A**) and percentage of embryos developed from in vitro settled ovules (**B**) for each group of crosses (characterized by female’s parent middle (M), middle-late (ML), and late (L) ripening time) at different sampling times (43–47 DAP, 48–52 DAP, 53–57 DAP, 58–62 DAP). For each sampling time and group of crosses, mean value (X), quartiles (-), and outstanding values (°) are reported; these data are condensed in X when there is only one sample.

**Figure 3 plants-12-03469-f003:**
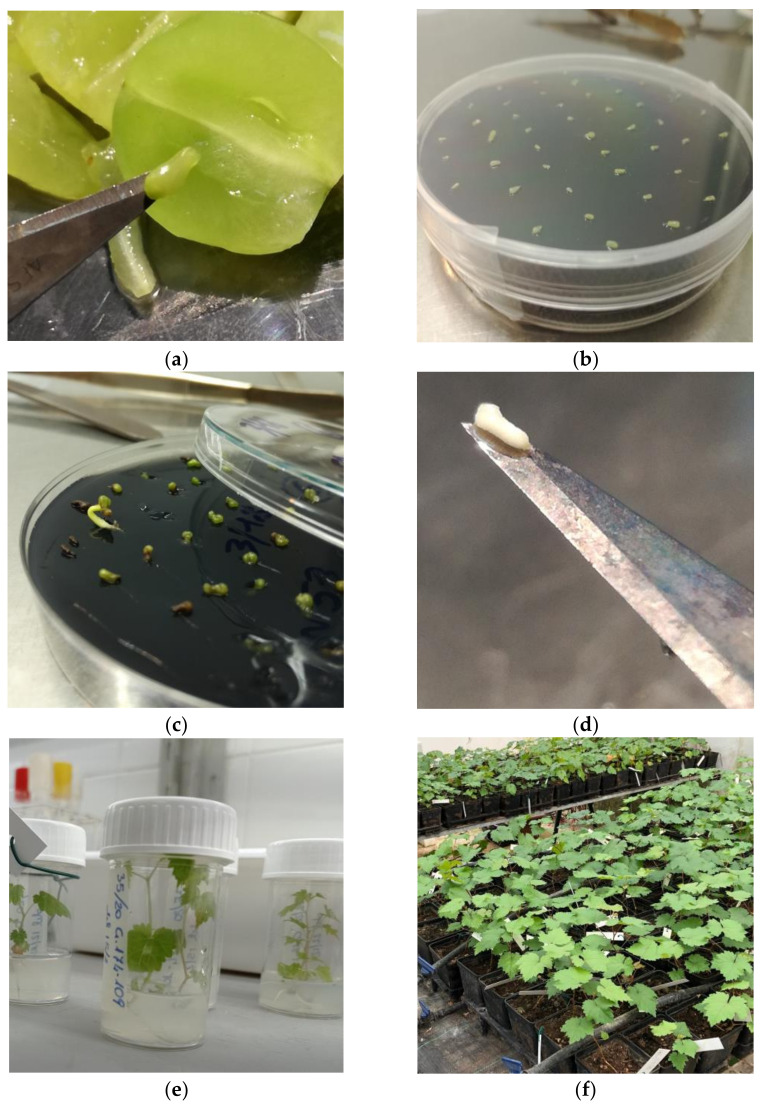
(**a**) Berry dissection for ovule extraction; (**b**) settling of ovules in Petri dishes on embryo growing medium; (**c**) spontaneous growth from ovules settled on embryo growth medium; (**d**) embryo extracted manually with a scalpel from mature ovule; (**e**) grown embryos, settled on S.O.R. medium; (**f**) acclimatized plants in pots located in screenhouses.

**Table 1 plants-12-03469-t001:** Data on plant material used in each cross and on the obtained embryos.

Crossed Cultivars (♀ × ♂)	Mean Number per Bunch ^a^			No. of Ovules per Berry	Developed Embryos (%)
	Berries	Ovules	Embryos		
TS01 × TS04	43.80 ± 10.58	97.40 ± 24.03	2.22 ± 0.20	3.90 ± 2.32	4.00 ± 1.40
TS02 × TS04	34.1 ± 6.0	58.9 ± 11.8	9.1 ± 1.9	1.7 ± 0.1	15.5 ± 2.5
TS04 × TS19	37.0 ± 4.3	88.0 ± 11.0	23.4 ± 6.7	2.4 ± 0.2	26.6 ± 6.3
TS04 × TS20	50.7 ± 9.5	111.4 ± 22.0	19.0 ± 5.6	2.2 ± 0.2	17.1 ± 3.6
TS04 × TS24	200.7 ± 25.4	442.3 ± 51.7	3.4 ± 0.7	2.2 ± 0.1	0.8 ± 0.2
TS05 × TS27	77.4 ± 8.0	154.6 ± 20.1	28.3 ± 4.0	2.0 ± 0.1	18.3 ± 1.8
TS06 × TS20	75.2 ± 15.7	140.7 ± 35.1	10.2 ± 3.2	1.9 ± 0.2	7.2 ± 2.5
TS06 × TS24	71.5 ± 16.8	177.8 ± 41.0	29.8 ± 9.7	2.5 ± 0.3	16.7 ± 3.4
TS07 × TS08	71.0 ± 8.5	138.8 ± 13.6	4.8 ± 1.7	2.0 ± 0.2	3.5 ± 1.4
TS09 × TS04	88.6 ± 19.0	211.9 ± 49.9	13.9 ± 3.7	2.4 ± 0.1	6.6 ± 1.0
TS09 × TS10	100.5 ± 16.8	240.9 ± 39.5	25.1 ± 3.5	2.4 ± 0.2	10.4 ± 1.9
TS09 × TS13	87.2 ± 8.3	204.2 ± 21.3	38.5 ± 5.1	2.3 ± 0.1	18.9 ± 1.3
TS09 × TS34	102.5 ± 13.9	213.0 ± 43.9	16.5 ± 5.3	2.1 ± 0.2	7.8 ± 2.0
TS11 × TS24	92.8 ± 18.7	197.5 ± 39.4	21.7 ± 7.3	2.1 ± 0.3	11.0 ± 2.7
TS12 × TS24	67.2 ± 10.6	111.4 ± 16.7	7.5 ± 2.8	1.7 ± 0.2	6.7 ± 2.4
TS13 × TS24	67.3 ± 8.2	138.1 ± 24.2	19.1 ± 4.5	2.1 ± 0.1	13.8 ± 3.8
TS13 × TS24	44.7 ± 5.5	90.4 ± 12.3	10.3 ± 1.6	2.0 ± 0.1	11.4 ± 1.5
TS13 × TS27	215.3 ± 22.2	389.9 ± 59.2	14.8 ± 2.3	1.8 ± 0.2	3.8 ± 0.8
TS14 × TS21	79.1 ± 15.5	224.8 ± 48.8	8.9 ± 3.3	2.8 ± 0.2	4.0 ± 1.1
TS15 × TS13	120.9 ± 11.3	309.8 ± 29.4	76.9 ± 10.5	2.6 ± 0.1	24.8 ± 3.0
TS15 × TS32	111.8 ± 31.1	200.4 ± 61.3	30.0 ± 7.4	1.8 ± 0.2	15.0 ± 4.9
TS17 × TS27	89.1 ± 9.3	172.8 ± 19.3	53.6 ± 6.8	1.9 ± 0.1	31.0 ± 2.5
TS22 × TS16	192.4 ± 20.0	342.7 ± 27.7	121.1 ± 13.8	1.8 ± 0.1	35.4 ± 4.0
TS23 × TS24	68.0 ± 9.3	169.5 ± 24.4	29.1 ± 6.4	2.5 ± 0.1	17.2 ± 2.9
TS23 × TS28	104.2 ± 12.9	213.9 ± 32.4	45.8 ± 6.3	2.1 ± 0.2	21.4 ± 4.2
TS24 × TS13	47.8 ± 21.4	107.0 ± 47.3	34.8 ± 14.3	2.2 ± 0.1	32.5 ± 13.6
TS25 × TS13	99.7 ± 9.5	158.9 ± 49.4	2.4 ± 0.8	1.6 ± 0.3	1.5 ± 2.3
TS25 × TS18	107.3 ± 15.1	228.1 ± 32.9	3.0 ± 1.1	2.1 ± 0.2	1.3 ± 0.6
TS25 × TS39	93.2 ± 14.4	128.0 ± 40.9	3.0 ± 1.4	1.4 ± 0.4	2.3 ± 1.6
TS27 × TS03	73.3 ± 4.9	243.8 ± 17.0	23.3 ± 2.7	3.3 ± 0.1	9.6 ± 1.5
TS27 × TS21	74.6 ± 6.9	170.4 ± 19.0	15.8 ± 3.1	2.3 ± 0.1	9.3 ± 2.0
TS27 × TS31	94.0 ± 10.8	236.4 ± 22.5	19.4 ± 6.1	2.5 ± 0.1	8.2 ± 3.4
TS27 × TS32	74.4 ± 3.7	239.3 ± 15.5	25.0 ± 3.0	3.2 ± 0.1	10.5 ± 1.1
TS27 × TS32	83.0 ± 9.4	156.8 ± 20.5	18.3 ± 3.8	1.9 ± 0.1	11.7 ± 1.9
TS28 × TS04	48.0 ± 8.5	56.8 ± 12.7	10.3 ± 4.0	1.2 ± 0.2	18.2 ± 3.9
TS28 × TS13	38.8 ± 5.8	40.0 ± 8.7	5.6 ± 1.5	1.0 ± 0.1	14.1 ± 5.8
TS29 × TS38	72.0 ± 7.8	143.6 ± 21.1	59.0 ± 9.4	2.0 ± 0.2	41.1 ± 4.8
TS30 × TS13	71.0 ± 13.0	182.5 ± 37.3	16.5 ± 2.9	2.6 ± 0.2	9.0 ± 0.5
TS33 × TS19	113.9 ± 10.8	270.8 ± 36.7	32.9 ± 9.0	2.4 ± 0.2	12.2 ± 3.2
TS33 × TS28	117.5 ± 22.8	279.8 ± 44.7	55.0 ± 15.7	2.4 ± 0.3	19.7 ± 4.6
TS35 × TS03	64.1 ± 10.3	186.1 ± 34.8	56.2 ± 15.5	2.9 ± 0.7	30.2 ± 4.1
TS36 × TS26	27.0 ± 3.6	55.6 ± 10.8	2.2 ± 0.6	2.1 ± 0.2	4.0 ± 0.6
TS37 × TS19	89.5 ± 6.9	204.1 ± 22.7	24.1 ± 4.1	2.3 ± 0.1	11.8 ± 2.5

^a^ Mean values ± Standard Error.

**Table 2 plants-12-03469-t002:** Efficiency of the protocol in terms of acclimatized plantlets.

Crossed Cultivars (♀ × ♂)	Ovules (No.)	Embryos (No.)	Plantlets (No.)	Embryos Acclimatized (%) *	Ovules Developed (%) **
TS01 × TS04	974	39	15	38.5	1.5
TS02 × TS04	589	91	33	36.3	5.6
TS04 × TS19	440	117	25	21.4	5.7
TS04 × TS20	780	133	17	12.8	2.2
TS04 × TS24	7961	61	21	34.4	0.3
TS05 × TS27	3092	565	232	41.1	7.5
TS06 × TS20	844	61	31	50.8	3.7
TS06 × TS24	711	119	64	53.8	9.0
TS07 × TS08	833	29	6	20.7	0.7
TS09 × TS04	1695	111	57	51.4	3.4
TS09 × TS10	2409	251	143	57.0	5.9
TS09 × TS13	4083	770	165	21.4	4.0
TS09 × TS34	852	66	27	40.9	3.2
TS11 × TS24	1185	130	43	33.1	3.6
TS12 × TS24	1225	82	14	17.1	1.1
TS13 × TS24	2485	344	88	25.6	3.5
TS13 × TS24	1266	144	17	11.8	1.3
TS13 × TS27	3509	133	54	40.6	1.5
TS14 × TS21	1798	71	33	46.5	1.8
TS15 × TS13	5577	1384	292	21.1	5.2
TS15 × TS32	1002	150	15	10.0	1.5
TS17 × TS27	3111	965	487	50.5	15.7
TS22 × TS16	2399	848	401	47.3	16.7
TS23 × TS24	1864	320	161	50.3	8.6
TS23 × TS28	2353	504	230	45.6	9.8
TS24 × TS13	428	139	42	30.2	9.8
TS25 × TS13	1112	30	16	53.3	1.4
TS25 × TS18	1825	24	12	50.0	0.7
TS25 × TS39	768	18	2	11.1	0.3
TS27 × TS03	6095	582	216	37.1	3.5
TS27 × TS21	1704	158	37	23.4	2.2
TS27 × TS31	1891	155	47	30.3	2.5
TS27 × TS32	7659	800	267	33.4	3.5
TS27 × TS32	1411	165	39	23.6	2.7
TS28 × TS04	341	62	29	46.8	8.5
TS28 × TS13	320	45	19	42.2	5.9
TS29 × TS38	1436	590	295	50.0	20.5
TS30 × TS13	730	66	22	33.3	3.0
TS33 × TS19	2979	362	203	56.1	6.8
TS33 × TS28	1679	330	206	62.4	12.3
TS35 × TS03	1675	506	249	49.2	14.9
TS36 × TS26	278	11	5	45.5	1.8
TS37 × TS19	3469	410	59	14.4	1.7

* % of developed embryos: the ratio was calculated by multiplying by 100 the ratio plantlets/embryos; ** % of developed ovules: the ratio was calculated by multiplying by 100 the ratio plantlets/ovules.

**Table 3 plants-12-03469-t003:** Seedless cultivars used in the breeding program and their main features.

Cross Code	Parents of Crossed Cultivar	Ripening Time	Main Features
TS01	hybrid Early gold × Sophia seedless	Middle	White
TS02	hybrid Pink muskat × Midnight beauty	Middle	Rose
TS03	Ribier × Black monuka	Early	Black
TS04	Sun world seedling × Sugarthirtone	Middle late	White
TS05	Fresno × Fresno	Late	White
TS06	Vitis interspecific crossing	Late	Black
TS07	Gold × Q 25-6 F2	Middle	White
TS08	Vitis interspecific crossing	Middle	White, aromatic
TS09	Eperor × Fresno	Late	Red
TS10	Emperor × Dawn seedless	Middle	Red
TS11	Gargiulo × ((Red malaga × Tifafihi Ahmer) × (Muscat of Alexandria × Sultanina))	Middle	Red
TS12	Local variety	Middle	Hybrid White, aromatic (foxy)
TS13	Local variety	Middle	Hybrid Black, aromatic (foxy), resistance to *E. necator* and *P. viticola*
TS14	Red Globe × Princess	Early	White
TS15	Emperor × Sultana moscata	Middle late	Red
TS16	Local variety	Late	Hybrid, White, resistance to *E. necator* and *P. viticola*
TS17	Emperor × Dawn seedless-	Middle	White, muscat flavor
TS18	Red Globe × Princess	Middle	Red
TS19	Unknown	Middle late	White
TS20	Sun World seedling × Fantasie seedless	Middle late	Black
TS21	Chasselas × Ahmeur Bou Ahmeur	Early	White, muscat flavor
TS22	Unknown	Middle	Red, muscat flavor
TS23	Datal × Centennial seedless	Middle late	White
TS24	Black Monukka × Sugarfive	Middle	Red, muscat flavor
TS25	Sun World seedling × Sun World seedling	Late	Red
TS26	Unknown	Early	White
TS27	Red Globe × Sun World seedling	Middle	White, muscat flavor
TS28	IFG × IFG	Middle	White
TS29	Fresno × Fresno	Early	Red
TS30	Princess × Regal seedless	Middle	White
TS31	Cardinal × unknown	Late	White
TS32	Cardinal × Kishmish Rozovyi	Late	Rose
TS33	Red Globe × Princess	Middle late	Red
TS34	USDA Selection × Princess	Middle late	White
TS35	Autum Royal seedless × unknown	Early	Red
TS36	IFG × IFG	Early	Red
TS37	Red Globe × Princess	Middle late	Red
TS38	Red Globe × Princess	Early	White, muscat flavor
TS39	Unknown	Late	Black

**Table 4 plants-12-03469-t004:** Numbers of sampled bunches and berries and in vitro settled ovules per cross.

Crossed Cultivars		Pollination Day	No. Bunches	No. Berries	No. Ovules
♀	♂				
TS01	TS04	21 May 2021	10	438	974
TS02	TS04	24 May 2021	10	341	589
TS04	TS19	5 June 2019	5	185	440
TS04	TS20	5 June 2019	7	355	780
TS04	TS24	22 May 2018	18	3612	7961
TS05	TS27	21 May 2020	20	1547	3092
TS06	TS20	22 May 2017	6	451	844
TS06	TS24	22 May 2017	4	286	711
TS07	TS08	11 June 2019	6	426	833
TS09	TS04	25 May 2021	8	709	1695
TS09	TS10	15 May 2017	10	1005	2409
TS09	TS13	20 May 2020	20	1743	4083
TS09	TS34	25 May 2021	4	410	852
TS11	TS24	25 May 2018	6	557	1185
TS12	TS24	1 June 2019	11	739	1225
TS13	TS24	22 May 2018	18	1212	2485
TS13	TS24	3 June 2019	14	626	1266
TS13	TS27	11 May 2017	9	1938	3509
TS14	TS21	12 May 2021	8	633	1798
TS15	TS13	8 June 2020	18	2176	5577
TS15	TS32	11 June 2019	5	559	1002
TS17	TS27	15 May 2018	18	1603	3111
TS22	TS16	29 May 2017	7	1347	2399
TS23	TS24	23 May 2017	11	748	1864
TS23	TS28	23 May 2017	11	1146	2353
TS24	TS13	5 June 2019	4	191	428
TS25	TS13	31 May 2021	7	698	1112
TS25	TS18	24 May 2017	8	858	1825
TS25	TS39	31 May 2021	6	559	768
TS27	TS03	11 May 2018	25	1832	6095
TS27	TS31	23 May 2019	8	752	1891
TS27	TS21	25 May 2019	10	746	1704
TS27	TS32	11 May 2018	32	2382	7659
TS27	TS32	23 May 2019	9	747	1411
TS28	TS04	27 May 2021	6	288	341
TS28	TS13	27 May 2021	8	310	320
TS29	TS38	20 May 2020	10	720	1436
TS30	TS13	28 May 2021	4	284	730
TS33	TS19	26 May 2017	11	1253	2979
TS33	TS28	26 May 2017	6	705	1679
TS35	TS03	10 May 2021	9	577	1675
TS36	TS26	11 May 2021	5	135	278
TS37	TS19	5 June 2019	17	1522	3469

**Table 5 plants-12-03469-t005:** Numbers of bunches, berries, and ovules analyzed per cross at different sampling time points.

Ripening Time *	Crossed Cultivar (♀ × ♂)	Samples (No.)	Sampling Time (Days after Pollination)			
			43–47	48–52	53–57	58–62
M	TS01 × TS04	Bunches	4	4	n.a.**	2
		Berries	285	104	n.a.	49
		Ovules	649	205	n.a.	120
M	TS02 × TS04	Bunches	2	4	n.a.	4
		Berries	62	146	n.a.	133
		Ovules	115	233	n.a.	241
M	TS11 × TS24	Bunches	n.a.	3	3	n.a.
		Berries	n.a.	232	325	n.a.
		Ovules	n.a.	527	658	n.a.
M	TS12 × TS24	Bunches	4	7	n.a.	n.a.
		Berries	349	390	n.a.	n.a.
		Ovules	633	592	n.a.	n.a.
M	TS13 × TS24	Bunches	7	11	n.a.	n.a.
		Berries	425	787	n.a.	n.a.
		Ovules	860	1625	n.a.	n.a.
M	TS13 × TS24	Bunches	6	8	n.a.	n.a.
		Berries	315	311	n.a.	n.a.
		Ovules	644	622	n.a.	n.a.
M	TS13 × TS27	Bunches	n.a.	5	4	n.a.
		Berries	n.a.	1157	781	n.a.
		Ovules	n.a.	1904	1605	n.a.
M	TS17 × TS27	Bunches	6	8	n.a.	3
		Berries	558	724	n.a.	321
		Ovules	1048	1462	n.a.	601
M	TS27 × TS03	Bunches	n.a.	10	10	5
		Berries	n.a.	754	751	327
		Ovules	n.a.	2610	2527	958
M	TS27 × TS21	Bunches	6	4	n.a.	n.a.
		Berries	1095	266	n.a.	n.a.
		Ovules	2337	622	n.a.	n.a.
M	TS27 × TS32	Bunches	n.a.	11	12	9
		Berries	n.a.	674	930	778
		Ovules	n.a.	2015	3113	2531
M	TS28 × TS04	Bunches	3	n.a.	n.a.	3
		Berries	119	n.a.	n.a.	169
		Ovules	141	n.a.	n.a.	200
M	TS28 × TS13	Bunches	3	n.a.	n.a.	5
		Berries	156	n.a.	n.a.	154
		Ovules	174	n.a.	n.a.	146
ML	TS04 × TS24	Bunches	6	7	5	n.a.
		Berries	1004	1490	1118	n.a.
		Ovules	2295	3845	1821	n.a.
ML	TS23 × TS24	Bunches	n.a.	7	4	n.a.
		Berries	n.a.	407	341	n.a.
		Ovules	n.a.	1032	832	n.a.
ML	TS23 × TS28	Bunches	n.a.	7	4	n.a.
		Berries	n.a.	659	487	n.a.
		Ovules	n.a.	1626	727	n.a.
ML	TS33 × TS19	Bunches	n.a.	6	n.a.	5
		Berries	n.a.	629	n.a.	624
		Ovules	n.a.	1646	n.a.	1333
ML	TS33 × TS28	Bunches	n.a.	4	n.a.	2
		Berries	n.a.	403	n.a.	302
		Ovules	n.a.	929	n.a.	750
ML	TS37 × TS19	Bunches	n.a.	8	n.a.	9
		Berries	n.a.	746	n.a.	776
		Ovules	n.a.	1741	n.a.	1728
L	TS05 × TS27	Bunches	n.a.	8	12	n.a.
		Berries	n.a.	808	739	n.a.
		Ovules	n.a.	1810	1282	n.a.
L	TS09 × TS04	Bunches	5	n.a.	3	n.a.
		Berries	464	n.a.	245	n.a.
		Ovules	1154	n.a.	541	n.a.
L	TS09 × TS10	Bunches	n.a.	8	n.a.	2
		Berries	n.a.	842	n.a.	163
		Ovules	n.a.	2105	n.a.	304
L	TS09 × TS13	Bunches	n.a.	10	n.a.	10
		Berries	n.a.	963	n.a.	780
		Ovules	n.a.	2176	n.a.	1907
L	TS25 × TS13	Bunches	n.a.	2	n.a.	5
		Berries	n.a.	214	n.a.	484
		Ovules	n.a.	442	n.a.	670
L	TS25 × TS18	Bunches	n.a.	3	5	n.a.
		Berries	n.a.	237	621	n.a.
		Ovules	n.a.	442	1383	n.a.
L	TS25 × TS39	Bunches	n.a.	2	n.a.	4
		Berries	n.a.	224	n.a.	335
		Ovules	n.a.	266	n.a.	502

* Ripening time: M: middle; ML: middle-late; L: late; **: n.a.: not analyzed.

## Data Availability

All data concerning the identity of parental varieties are property of the Rete Italian Variety and therefore unavailable for commercial rights reasons.

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
