# Peer review of "A Large-Scale Validation of an Improved Embryo-Rescue Protocol for the Obtainment of New Table-Grape Seedless Genotypes"

_plants, 2023, doi:10.3390/plants12193469_

Round 1

Reviewer 1 Report

The manuscript "A large-scale validation of an improved embryo-rescue protocol for the obtainment of new table-grape seedless genotypes " represents good work. I am recommending this manuscript for publication with some modifications as follows:

1. A similar work was published in 2013 under the title “Breeding new seedless grape by means of in vitro embryo rescue” The authors need to add more knowledge to this area.

2. The authors need to add more molecular work on the level of DNA for example Marker-assisted selection, and PCR assays to confirm the success of their work.

3. The language or sentence formations need to be addressed carefully for better understanding.

The language or sentence formations need to be addressed carefully for better understanding.

Reviewer 2 Report

The manuscript entitled 'A large-scale validation of an improved embryo-rescue protocol for the obtainment of new table-grape seedless genotypes' focuses on perfecting an embryo-rescue strategy for the cultivation of embryos derived from various crossings involving seedless parents that were specifically chosen for their desirable traits. The research issue under consideration is relevant and aligns with the preferences of customers who seek to consume grapes without seeds. The results emphasize the need of a meticulous selection of parental genotypes in breeding crosses.

The manuscript exhibits a good level of writing proficiency, effectively presenting arguments in support of the ongoing research and highlighting significant findings that hold relevance for future breeding initiatives.

In order to succeed publication, I strongly recommend the authors to update the reference list: out of the total number of 52 sources, barely 15 have been published during the recent decade. There exist outdated references (1933, 1936, 196x etc.) that, despite their inherent value, raise doubts regarding the current validity of information. Has there been a lack of recent research on this particular subject matter in the last 90 years? Please check and provide more up-to-date references. Also, I propose that the authors include in Materials and Methods section, details regarding the software used to perform the processing of statistical data.

Overall, it is a clear and interesting study. I extend my well wishes to the authors in their future research efforts.

Round 2

Reviewer 1 Report

I am thankful to the authors for providing the required modifications to the manuscript "A large-scale validation of an improved embryo-rescue protocol for the obtainment of new table-grape seedless genotypes ". I am recommending this manuscript for publication.

Regards.

Ahmed Darwish

Author Response

Thank you for your suggestions.

Regards